# Production of Astaxanthin Using CBFD1/HFBD1 from *Adonis aestivalis* and the Isopentenol Utilization Pathway in *Escherichia coli*

**DOI:** 10.3390/bioengineering10091033

**Published:** 2023-09-01

**Authors:** Jared H. Roth, Valerie C. A. Ward

**Affiliations:** Department of Chemical Engineering, University of Waterloo, Waterloo, ON N2L 3G1, Canada

**Keywords:** carotenoids, hydrophobic products, *Adonis aestivalis*, *Escherichia coli*, isopentenol utilization pathway (IUP)

## Abstract

Astaxanthin is a powerful antioxidant and is used extensively as an animal feed additive and nutraceutical product. Here, we report the use of the β-carotene hydroxylase (CBFD1) and the β-carotene ketolase (HBFD1) from *Adonis aestivalis*, a flowering plant, to produce astaxanthin in *E. coli* equipped with the *P. agglomerans* β-carotene pathway and an over-expressed 4-methylerythritol-phosphate (MEP) pathway or the isopentenol utilization pathway (IUP). Introduction of the over-expressed MEP pathway and the IUP resulted in a 3.2-fold higher carotenoid content in LB media at 36 h post-induction compared to the strain containing only the endogenous MEP. However, in M9 minimal media, the IUP pathway dramatically outperformed the over-expressed MEP pathway with an 11-fold increase in total carotenoids produced. The final construct split the large operon into two smaller operons, both with a T7 promoter. This resulted in slightly lower productivity (70.0 ± 8.1 µg/g·h vs. 53.5 ± 3.8 µg/g·h) compared to the original constructs but resulted in the highest proportion of astaxanthin in the extracted carotenoids (73.5 ± 0.2%).

## 1. Introduction

Astaxanthin is a red carotenoid and a highly valuable antioxidant used in the pharmaceutical, cosmetics, nutraceutical, and aquaculture industries [1,2]. It is also used as a nutraceutical for preventing diseases caused by oxidative stresses, such as cataracts disease, various cancers, Parkinson’s disease and Alzheimer’s disease [3]. It has significant applications in fish farming, where astaxanthin is included in the feed of salmon, trout, and shrimp to brighten the colour of their meat [1]. Astaxanthin is naturally synthesized by several species of algae and fungi [4,5,6]. The majority of commercial astaxanthin production is by chemical synthesis. Unfortunately, the resulting product is a mix of stereoisomers [5], and there is substantial consumer desire for biologically produced astaxanthin containing the single isomer produced by biosynthesis.

Most industrial biological production of astaxanthin uses the microalgae species *Haematococcus pluvialis* in outdoor photobioreactors, which can accumulate up to 50 mg/g of astaxanthin, the highest reported specific yield for this compound [7,8]. However, the life cycle of this algae species is slow and complex, and astaxanthin production must be induced by using some sort of stressor, typically high-intensity light that is difficult to scale up [9,10]. The production of astaxanthin in heterologous hosts has also been studied extensively for the last 20 years, with the industrial workhorses *Escherichia coli*, *Saccharomyces cerevisiae* and *Yarrowia lipolytica* being the most popular species for heterologous carotenoid production [11]. Astaxanthin produced in *E. coli* is readily extracted due to its simple cell walls, and as a bacterium, its cultivation is straightforward [12], making astaxanthin production in *E. coli* an attractive host. While specific yields are typically 10-fold lower than those achieved with *H. pluvialis*, *E. coli* can be more readily cultivated into much higher cell densities in a much shorter period than *H. pluvialis*.

*E. coli* possesses the biosynthetic pathway needed to produce up to sesquiterpenoids from farnesyl pyrophosphate (FPP) using its native methylerythritol-4-phosphate (MEP) pathway (Figure 1) [13]. To produce the first coloured carotenoid, lycopene, the genes *crtE*, *crtI*, and *crtB*, typically sourced from *Pantoea agglomerans* or *anantis*, must be expressed [11]. In order to achieve higher carotenoid yields in *E. coli,* many studies co-express the heterologous mevalonate pathway (MVA) from eukaryotes [14], and recently, the artificial isoprenoid biosynthesis pathway called the isopentenol utilization pathway (IUP) has been used to achieve very high lycopene yields [15].

Carotenoid production is highly conserved amongst diverse species until lycopene. Enzymes used to produce β-carotene, the precursor to xanthophylls, and the remaining steps to form astaxanthin can vary significantly between different phyla (Figure 1). However, today, the majority of studies looking to produce astaxanthin in *E. coli* have expressed the bacterial *crtY* from *Pantoea* sp. or *lycB*, a plant lycopene β-cyclase to produce β-carotene [11]. In order to produce astaxanthin, β-carotene then needs to be oxidized by both ketolases and hydroxylases. Natural bacterial producers use CrtZ and CrtW, microalgae use CrtO/BKT and CrtR/Chyb, some fungi use a bifunctional CrtS (paired with cytochrome P_450_ reductase CrtR), and HBDF and CBFD are used in flower plants [16]. The biosynthesis routes of each pair of enzymes produce different intermediate species. While a multitude of bacterial ketolases and hydroxylases have been used in metabolic engineering of astaxanthin in *E. coli*, there are no studies to date exploring the use of carotenoid-β-ring 4-dehydrogenase (CBFD1) and carotenoid-4-hydroxy-β-ring 4-dehydrogense (HBFD1), sourced from the flowering plant *Adonis aestivalis* with an over-expressed isoprenoid pathway [16]. Therefore, the major goal of this work was to evaluate the productivity of *E. coli* strains with over-expressed MEP and IUP pathways when using the CBFD1/HBFD1 pathway for astaxanthin production.

In this work, we combine the astaxanthin biosynthesis pathway of *A. aestivalis* and the carotenoid pathway of *P. agglomerans* with either an upregulated endogenous MEP or the artificial IUP biosynthesis pathway for increased IPP/DMAPP production in *E. coli* to produce astaxanthin. The relative portion of astaxanthin and other carotenoid intermediates was determined by HPLC and was highly dependent on the construction of the plasmids used. Total productivity was highly dependent on the cultivation media for different upstream pathways for the overproduction of IPP and DMAPP precursors.

## 2. Materials and Methods

### 2.1. Strains, Plasmids and Genes

*E. coli* K12, MG1655 (DE3) was used as the host for all the astaxanthin expression studies in this work. MG1655(DE3)-trcMEP was gifted from the Stephanopoulos lab (MIT, MA, USA) and has four MEP genes under the control of a lac inducible trc promoter inserted into the chromosome near the arabinose operon [17]. NEB-5α was used for routine cloning purposes. The genotypes of these strains and plasmids are available in Table 1. Origin of genes and their accession numbers are listed in Table A1. The genes from the astaxanthin production pathway were amplified according to the protocol given by the NEB Phusion PCR kit and extracted from a 1% agarose gel. The genes for *ggpps*, *crtB*, *crtI* and *idi* were sourced from p5T7-lycipi-ggpps [15], which was used as a backbone for the synthesis of p5T7-Astaipi. The *crtY* gene was sourced from pAC-BETAipi, and *cbfd* and *hbfd* were sourced from pCBFD1, both of which were purchased from Addgene (Watertown, MA, USA) (plasmid #53277 and #53364). To over-express *ispA*, the gene was added to p5T7-lycipi-ggpps from p5T7-Ispa-ads to create p5T7-lycipi-ispA. Using two steps, a T7 promoter, terminator, and lac operator (lacIQ) were added to pAC-BETAipi to make pACT7-CBFD1, and then pAC-ASTA was created from this plasmid to house the rest of the genes in the pathway (*cbfd1*, *crtY*, *hbfd1*) under a single T7 promoter. A summary of each construct is shown in Figure 2. All genes expressed in operons have their own ribosome binding site (RBS) except for pAC-BETAipi and pCBFD1, which were obtained from Addgene and used as is. All plasmid sequences are available by request.

A list of primers used in this work can be found in Table A2. The fragments were ligated using NEB Hi-Fi assembly master mix and transformed into chemically competent NEB-5α cells using heat shock. Colony PCR was performed using Taq DNA polymerase (New England Biolabs, MA, USA) and standard buffer to identify positive transformants, and the plasmid was isolated and sequenced to confirm the correct assembly. The plasmids were electroporated into electrocompetent cells in cuvettes with a 1 mm gap (1.8 kV, 25 μF capacitance) and grown on LB plates with the appropriate antibiotics to make the strains listed in Table 1.

### 2.2. Cultivation Conditions

All media were prepared according to the descriptions below and autoclaved or filter-sterilized prior to use. Antibiotics and inducer stocks were made at 1000× concentration, filtered and stored at −20 °C. Final concentrations of antibiotics were Kn (50 μg/mL), Ap (50 μg/mL), and Sp (50 μg/mL). Strains were cultivated in either LB media (10 g/L tryptone, 5 g/L yeast extract, 10 g/L NaCl) or M9 media containing 3.2 g/L glucose, 5 g/L KH_2_PO_4_, 1 g/L NH_4_Cl, 0.5 g/L NaCl, 6.78 g/L Na_2_HPO_4_, 100 μM CaCl_2_, 2 mM MgSO_4_, and 10 mL/L trace elements based on the formulation provided by Wolfe [19]. Strains were stored at −80 °C in glycerol stocks and revived on LB agar plates (1.5% agar), which were grown overnight at 37 °C. A single colony was inoculated into LB or M9 media and grown overnight to prepare an inoculum. For carotenoid production, strains were inoculated with 1% (*v*/*v*) of overnight culture and cultivated in triplicate in 50 mL of M9 media at 30 °C with shaking at 200 rpm. At an OD_600_ of 0.5, carotenoid production was induced using a final concentration of 25 mM isoprenol (IUP strains), 1 g/L arabinose (P_BAD_ strains), and 0.1 mM IPTG (P_T7_ strains) unless otherwise indicated. A list of the strains used in this study can be found in Table 1.

### 2.3. Carotenoid Extraction and UV/Vis Spectroscopy

For carotenoid quantification, two methods were used, total carotenoid determination using spectrophotometry or liquid chromatography combined with a diode array detector. In order to determine the carotenoid content, two 1 mL samples were taken from each flask at the indicated time after induction. Samples were stored in amber microtubes to prevent photodegradation. The cell pellet was collected by 12,000× *g* for 1 min. One pellet was lyophilized and weighed to obtain the cell dry weight. The other was extracted with 1 mL of 1:1 (v:v) ethanol-acetone solution. The samples were vortexed to mix and were incubated in the dark for 1 h at room temperature. The samples were centrifuged again at 12,000× *g* for 1 min, and 200 μL of the liquid phase was transferred to a 96-well plate, and absorbance was measured using a BioTek Synergy 4 (Agilent, CA, USA) plate reader at 475 nm. Astaxanthin was purchased from Santa Cruz Biotechnologies (Dallas, TX, USA) and used to create a standard curve and was used as a proxy for total carotenoids. Total carotenoids were calculated using the following equation:$$Total\;Carotenoids\;\left(\mu g/g\right)=\frac{Abs-blank}{0.0799\;mL/\mu g}÷dry\;cell\;weight\;(g/mL)$$

### 2.4. Carotenoid Characterization

Carotenoids were extracted as described above and analyzed using high-performance liquid chromatography (1260 Infinity II, Agilent, CA, USA) equipped with a C30 column (YMC Carotenoid column, 250 mm, 5 μm pore size). Mobile phase A consisted of 15:81:4 Methyl tert-Butyl Ether (MTBE):methanol:water by volume, and mobile phase B consisted of 81:15:4MTBE: methanol:water by volume. Using a flow rate of 1.0 mL/min at 20 °C, a linear elution gradient from 100% A to 100% B over 15 min was followed by 12 min of 100% B before returning to mobile phase A over 3 min. HPLC standards (astaxanthin, lycopene, β-carotene, zeaxanthin, and canthaxanthin) were purchased from Santa Cruz Biotechnology for identification of carotenoid retention times. Zeaxanthin was used to identify isozeaxanthin as this compound cannot be purchased, and these isomers are known to co-elute using C18 chromatography [20].

## 3. Results

To compare the effects of different upstream pathways on the production of astaxanthin in an existing system, pAC-BETAipi and pCBFD1 plasmids were transformed into MG1655 (DE3), MG16655 (DE3) with trcMEP operon inserted into the chromosome and co-transformed with the pSEVA228-pro4IUP plasmid resulting in strains ASTA 1, ASTA 2, and ASTA 3, respectively. Each strain was grown in LB media as well as M9 media, and the results are presented in Figure 3.

In complex media such as LB, the strains expressing an upregulated MEP pathway were the most productive for carotenoid production, resulting in a maximum carotenoid titre of 6.05 ± 0.95 mg/L at 36 h (Figure 3A). This was a 2.5-fold increase in carotenoid titre over the wild-type strain. The IUP expressing strain had a lower titre than the trcMEP strain, but both strains reached the same carotenoid content by 36 h (2.87 ± 0.58 and 2.87 ± 0.67 at 36 h, respectively). These results are explained by the higher cell density of the wild-type and trcMEP strains over the course of the cultivation. The IUP strain only reached half the cell density of the wild-type strain (1.37 ± 0.25 g/L vs. 2.67 ± 0.58 g/L, respectively). Interestingly, when grown in M9 media, a minimal glucose media, the IUP strain dramatically outperformed the trcMEP and wild-type strains, producing 11.3 ± 0.55 mg/L of total carotenoids. This was a 13-fold increase over the wild-type strain and an 11-fold increase over the trcMEP strain in M9 media. This is still approximately double the titre produced by the trcMEP strain in LB media. The type of media used also had an effect on when carotenoid production ceased. In LB media, the maximum carotenoid titre and content were reached by 36 h. However, in M9 media, production of carotenoids continued until 48 h in the IUP strain but ceased by 12 h in the wild-type MEP and trcMEP strains. These differences are likely due to the depletion of nutrients in LB/M9 for wild-type and trcMEP strains, which depend on glucose or amino acids from the media for precursors through the MEP pathway. In the IUP strain, biosynthesis of carotenoids could continue because of the exogenous isoprenol added to the media that is not used for central carbon metabolism or cell maintenance energy. In order to observe the role of the downstream operon structure in different media, a new plasmid was constructed (p5T7-Astaipi) with all of the genes necessary for astaxanthin production under the control of the T7 promoter. This was also constructed to reduce the metabolic burden of the IUP strain that required three plasmids for carotenoid production. CrtE was replaced with *ggpps* from *Taxus canadensis*, which was previously reported to increase lycopene production in *E. coli* [15], and a copy of *ispA* was added to increase FPP production. The results are presented in Figure 4. A similar trend was obtained using a different downstream plasmid with the trcMEP strain, resulting in higher titres in complex media (Figure 4A) and the IUP pathway, resulting in higher titres in the minimal media (Figure 4B). When compared to the original two plasmid system, the trcMEP titre was not significantly different at 36 h in LB media (6.05 ± 0.95 vs. 5.26 ± 0.49, *t*-test *p* > 0.01, *n* = 3) or M9 media (1.01 ± 0.11 vs. 0.99 ± 0.06, *t*-test *p* > 0.01, *n* = 3). However, the IUP strain had a 3.6-fold decrease in titre with the new single plasmid system. The isoprenoid pathways and the carotenoid pathways are known for their sensitivity to protein levels, and many studies have observed that precise balancing of proteins may be needed to achieve the best titres [21,22].

The operon of the p5T7-Astaipi plasmid contains seven coding sequences. Due to the length, the translation of genes near the end of the operon may be less frequent than those at the front, as placement in an operon is known to affect translational efficiency [23]. The plasmid used to make p5T7-Astaipi; p5T7-lycipi-ggpps has been reported as one of the fastest producers of lycopene in the literature. Therefore, the remaining *crtY*, *cbfd1*, and *hbfd1* genes were placed together in an operon controlled by the same T7 promoter, and a copy of *ispA* was added to p5T7-lyc-ggpps to create pAC-ASTA and p5T7-lycipi-ispA. The new plasmids were combined with the wild-type MEP, the trcMEP, and the IUP upstream pathways and grown in M9 media. The results are shown in Figure 4.

The results were expected to be similar to the previous single-operon system. The IUP strain has the same titre and carotenoid content as the previous plasmid system; they both peaked at 24 h with total carotenoid titres of 3.65 ± 0.39 mg/L (ASTA 9) and 3.42 ± 0.40 mg/L (ASTA 6), respectively (Figure 5). However, the new system with two operons performed better for the trcMEP and wild-type strains, increasing the titre 2.8-fold and 4.8-fold, respectively. HPLC analysis of the carotenoids produced in strains ASTA 3, 6, and 9 showed that astaxanthin was produced in all strains, although strains ASTA 6 and ASTA 9 produced significantly more than ASTA 3 (Figure A1). All strains contained some amount of unconverted carotenoid intermediates, with ASTA 3 producing mostly β-carotene.

When comparing the productivity of all strains in M9 media over the 48 h cultivation period, the IUP strains outperformed the endogenous MEP and the strain with an over-expressed MEP (Figure 6A). The first strain (ASTA 3) had the highest productivity of all of the strains, but only a small portion of the products was astaxanthin (Figure 6B). The first iteration of new plasmids placed the *cbfd1* under the T7 promoter instead of the arabinose promoter. This resulted in a greater portion of astaxanthin (34.6 ± 3.0% vs. 56.4 ± 0.8%) and a decrease in β-carotene production (40.7 ± 4.3% vs. 24.6 ± 0.5%). However, in strain ASTA 9, astaxanthin was the major product (73.5 ± 0.2%), and no canthaxanthin was detected.

## 4. Discussion

The astaxanthin β-carotene hydroxylase (CHY) and ketolase enzymes from *A. aestivialis* used in this work (CBFD1/HBFD1) have not previously been used in metabolic engineering efforts for xanthophyll production. A survey of the literature shows that almost all studies to date have focused on the use of CrtW and CrtZ from a limited number of bacterial species (Table 2), with a small number of studies employing the BKT enzyme from *C. reinhardtti*, and the CHY from the microalgae *H. pluvialis* [24]. Cunningham et al. (2011) [16] first reported the production of astaxanthin in *E. coli* using CBFD1 and HBFD1. However, the production of carotenoids and relative composition were not reported for this gene combination. The operon construction had a significant impact on the overall productivity of the strain, as did the combination of upstream isoprenoid and carotenoid pathways (Figure 3, Figure 4 and Figure 5). Interestingly, a striking difference in productivity was found for strain ASTA 3 in LB and M9 media.

The differences In carotenoid production between the ASTA 1–3 and ASTA 4–6 strains may be due to the different promoters used in each system. Minimal media supplemented with glucose activates catabolite repression, which can lead to lower transcription levels for certain promoters such as the trc promoter [25], which explains why carotenoid titre and content decreased in M9 media for the trcMEP strain (6-fold decrease in titre). However, the pro4 promoter is a synthetic promoter [26], which should not be affected by catabolite repression, but the titre was 3-fold higher in M9 than LB media. This could be a significant advantage for the IUP pathway as minimal salt-based media are inexpensive at large scale and may result in greater reproducibility. Currently, it is unknown why carotenoid production was significantly higher in the minimal media with glucose, as productivity is normally decreased in these types of media. Presumably, this is because the cell must dedicate greater resources towards de novo synthesis of nucleotides, amino acids, and vitamins that would be obtained from rich media ingredients such as yeast extract. However, there are many possible reasons for this difference, such as large changes in overall metabolic flux balance, isoprenol binding to peptides in the media through hydrogen bonding, changes in the rate of isoprenol evaporation from the media, or changes in gene expression levels in different media and to elucidate these differences will be the subject of a more extensive investigation.

**Table 2 bioengineering-10-01033-t002:** Summary of the carotenoid content and titres reported in the literature and the genes used in previous studies.

Isoprenoid Pathway	β-Carotene Biosynthesis Genes	β-Carotene Hydroxylase and Ketolase	Specific Yield	Astaxanthin Purity	Notes	Ref.
Artificial IUP pathway						
IUP	*crtEBIY—Pantoea agglomerans* *ggpps—Taxus canadensis*	*cbfd*, *hbfd—A. aestivalis*	3.91 mg/g 2.66 mg/g	34.6% 73.5%	Performance of IUP better in minimal media	This study
Endogenous MEP pathway						
MEP	*crtEBIY—Paracoccus haeundaensis*	*crtW—P. haeundaensis* *crtZ—P. haeundaensis*	0.4 mg/g	n.d.		[27]
MEP	*crtEBIY—Pantoea agglomerans*	*crtW—Anabaena variabilis* *crtZ—S. solfataricus*	0.3 mg/g	71%	Screened various β-carotene hydroxylases	[21]
MEP	*crtEBIY—P. agglomerans*	*crtW—Nostoc* sp. *crtZ—P. agglomerans*	1.99 mg/g	>90%	Screened various β-carotene ketolases	[28]
MEP	*crtEBIY—P. ananatis*	*crtW—Brevundimonas* sp. *crtZ—P. ananatis*	7.4 mg/g	96.6%		[29]
MEP	*crtEBIY—P. ananatis*	*crtW—Brevundimonas* sp. *crtZ—P. ananatis*	11.92 mg/g	n.d.	Increase ROS Changing morphology	[30]
MEP	*crtEBIY—P. ananatis*	*BKT—Chlamydomonas reinhardtii* *CHY—H. pluvialis*	4.30 mg/g	n.d.		[24]
MEP	*crtEBIY—P. ananatis*	*crtW—A. aurantiacum* *crtZ—P. ananatis*	8.3 mg/g	n.d.	Multiple promoters	[31]
MEP	*crtEBIY—P. ananatis*	*crtW—Brevundimonas* sp. *crtZ—Brevundimonas* sp.	0.58 mg/g	~60%	Fusion proteins	[32]
Engineered MEP pathway						
MEP + extensive host changes	*crtEBIY—P. ananatis*	*crtW—Brevundimonas* sp. *crtZ—P. ananatis*	~12 mg/g	n.d.	Used adaptive laboratory evolution to create mutant strains Shuffled strain with CRISPRi and CRISPRa	[33]
MEP + *E. coli idi*	*Ch(crtEBIY)—P. ananatis*	*Ch(crtZ)—P. ananatis* *Ch(crtW148)—N. punctiforme*	1.4 mg/g	>95%		[34]
MEP + *E. coli idi*	*Ggpps—Archaeoglobus fulgidus * *CrtBIY—A. aurantiacum*	*crtW—A. aurantiacum* *crtZ—A. aurantiacum*	1.25 mg/g	n.d.		[35]
MEP + *H. pluvialis idi + E. coli dxs*	*crtEBI—P. agglomerans* *LycB—Solanum lycopersicum*	*crtW—N. sphaeroides* *crtZ—P. ananatis*	5.8 mg/g	Majority	RBS optimization	[36]
MEP + *E. coli idi*, *ispA*	*crtE—P. ananatis* *crtIBY—A. aurantiacum*	*crtZW—A. aurantiacum*	1.4 mg/g	n.d.		[35]
MEP + *E. coli ispDF*	*crtEBIY—P. ananatis*	*crtZ—P. ananatis* *trBKT—Chlamydomonas reinhardtii*	7.12 mg/g	n.d.	Used fusion tags to solubilize truncated BKT	[12]
MEP + *E. coli idi*, *ispA*, *ispH*	*crtEYIB—P. haeundaensis*	*crtW—P. haeundaensis* *crtZ—P. haeundaensis*	1.2 mg/g	n.d.		[37]
MEP + *K. gwangalliensis ispCDEFGH*, *idi*	*crtEYIB—P. haeundaensis*	*crtW—P. haeundaensis* *crtZ—P. haeundaensis*	1.10 mg/g	~65%		Jeong 2018 microbial letters
Modified MEP and other pathways	*Ch(crtEYIB)—P. agglomerans*	*crtW—Brevundimonas* sp. *crtZ—P. agglomerans*	5.88 mg/g	99%	Mutant library of crtW was screened	[38]
Modified MEP and other pathways	*Ch(crtEYIB)—P. agglomerans*	*Ch(crtW-GlpF) Brevundimonas* sp. *Ch(crtZ-GlpF) P. agglomerans*	~0.28 mg/g (AX)	n.d.	Fusion to membrane protein glpF	[39]
Heterologous MVA pathway						
MEP + MVA	*Ch(crtEBIY)—P. ananatis* *Ch (crtY—*additional copy)—*P. ananatis*	*crtZ—P. ananatis (*2 copies) *crtW—P. ananatis* (2 copies) *crtZ—Paracoccus* sp. PC1 (1 copy)	2.9 mg/g	65%	Fed-batch fermentation	[14]
MEP + MVA	*crtEBI—P. ananatis* *crtY—P. agglomerans*	*crtW- Brevundimonas* sp. *crtZ—P. agglomernas*	6.6 mg/g (AX)	n.d.		[40]
MEP + MVA	*crtEBI—P. agglomerans* *crtY—P. ananatis*	CrtZ—*P. ananatis* LMG20103 crtW *Brevundimonas* sp. SD212	~6 mg/g *	~85%	RBS optimization Fed-batch fermentation with in situ product removal	[41]
MEP + MVA	*Ch(crtEBIY)—P. agglomerans*	*crtZ—Brevundimonas* sp. *SD212* *crtW—Paracoccus* sp. *N81106*	4.67 mg/g	N.d.	crtZ/W fusion proteins	[42]
MEP + MVA	*crtEYIB*	*crtW—P. agglomerans* *crtZ—P. agglomerans*	6.17 mg/g	32%	RBS optimization Expressed chaperones groES-groEL Fed-batch fermentation	[43]

* Estimated using the cell dry weight correlation of 0.33 g/L/OD_600_ for *E. coli.* Ch() represents chromosomal integration of the listed genes.

The accumulation of intermediate carotenoids in each strain also differed depending on the structure of the carotenoid operon(s). When *cbfd1* was moved from the arabinose promoter to a stronger T7 promoter and when the operon was split into two operons controlled by two separate T7 promoters, astaxanthin production increased. In strain ASTA 9, there was no accumulation of canthaxanthin, suggesting that HBFD1 might be the rate-limiting step in this strain. Likely, CBFD1 was the rate-limiting step in strain ASTA 3 as there was a significant amount of β-carotene accumulating in this strain (Figure 6 and Figure A1). Fusions of CrtW/Z have been shown to be an effective strategy for increasing the conversion of zeaxanthin to astaxanthin by localizing the subsequent enzyme near the site of product formation [32,42]. Similarly, fusion to membrane proteins for targeted localization also improved astaxanthin production [39]. Chou et al. (2019) also found multiple promoters enhanced the biosynthesis of astaxanthin by increasing the efficiency of β-carotene conversion compared to using a single-operon system [31]. From Table 2, it can be seen that the species or origin, copy number, promoter, and combination of upstream and downstream genes used to play a significant role in the overall productivity of the system. The highest astaxanthin content found to date was achieved in strains with changes to membrane morphology and higher reactive oxygen species (ROS) levels [30]. However, these strains also exhibited decreased cell growth. Perhaps using CRISPR interference (CRISPRi) in a two-stage process might allow higher astaxanthin production after the majority of cell growth has occurred. A summary of this study and the changes and improvements made are shown in Figure 7. 

## 5. Conclusions

The IUP pathway significantly increased carotenoid production in *E. coli* in minimal media rather than complex media. There was an 11-fold increase in carotenoid yield in M9 media compared to LB media. The genes *cbfd1/hbfd1* were capable of producing astaxanthin at a similar level to the CrtW/Z of bacterial origin. Similarly, the bottlenecks in the xanthophyll portion of the pathway were dependent on the promoters and operon organization of the carotenoid pathway genes and *cbfd1/hbfd1,* as seen in other reports. Future work elucidating the effect of growth media on overall productivity may provide insights that will improve astaxanthin production. Finally, future studies into possible combinations of CBFD1/HBFD1 and CrtW/Z enzymes with complementary specificities to alleviate possible bottlenecks in the xanthophyll portion of the pathway may be useful for increasing the proportion of astaxanthin produced without reducing the overall carotenoid productivity. 

## Figures and Tables

**Figure 1 bioengineering-10-01033-f001:**
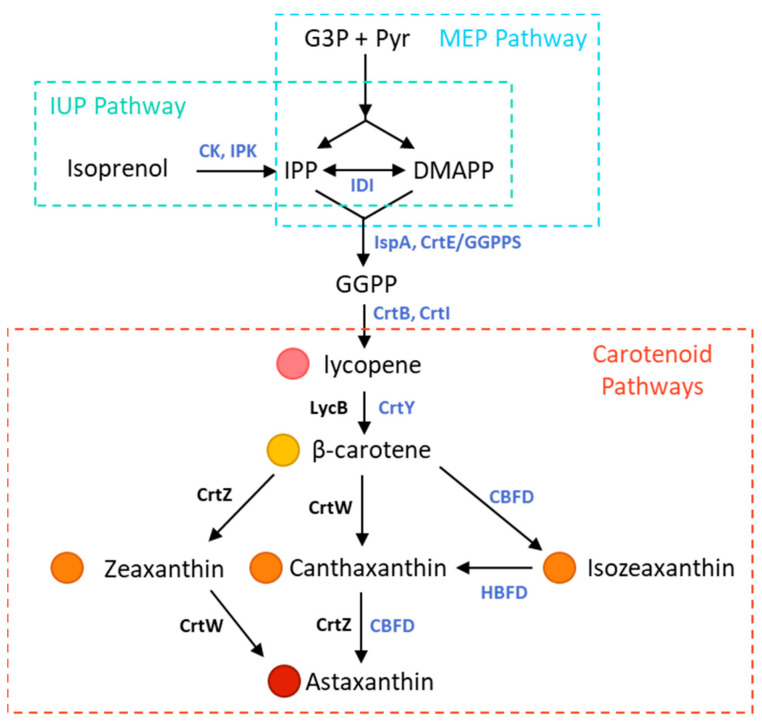
Biosynthesis pathways for heterologous production of astaxanthin in *E. coli*. *E. coli* possesses an endogenous MEP pathway starting from G3P and Pyr to form both IPP and DMAPP. The IUP uses the enzymes CK and IPK to produce IPP from isoprenol. DMAPP and IPP ratios are modulated by IDI. IspA and GGPP synthase convert IPP and DMAPP into GGPP. Two GGPP are converted into lycopene (pink) by CrtB and CrtI. β-carotene (yellow) is formed by cyclases CrtY or LycB. Bacterial ketolase CrtW and hydroxylase CrtZ produce astaxanthin (red) via isozeaxanthin (orange) and canthaxanthin (orange) intermediates. Flowering plants use CBFD and HBFD to produce astaxanthin via isozeaxanthin (orange). Enzymes used in this work are shown in blue. Modified from [11,16] **G3P**: glyceraldhyde-3-phoshpate; **Pyr:** pyruvate; **IPP:** isopentenyl diphosphate; **DMAPP:** dimethylallyl diphosphate; **FPP:** farnesyl diphosphate; **GGPP:** geranylgeranyl diphosphate; **CK:** choline kinase; **IPK:** isopentenyl phosphate kinase; **IDI:** isopentenyl diphosphate isomerase; **IspA:** FPP synthase; **CrtE/GGPPS**: GGPP synthase.

**Figure 2 bioengineering-10-01033-f002:**
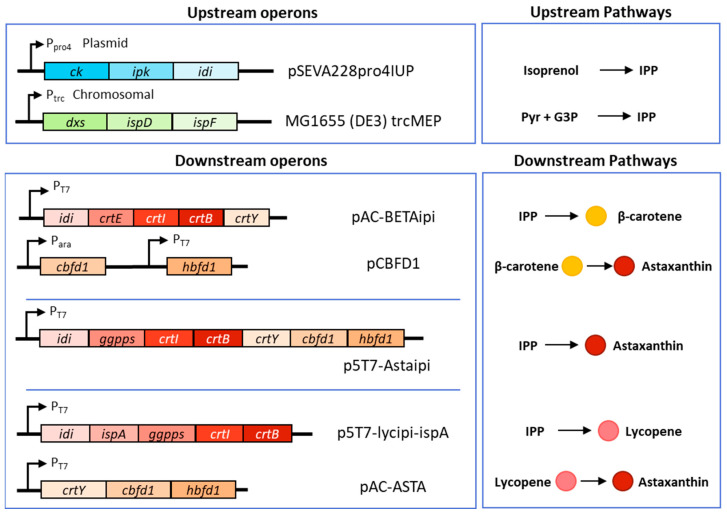
Plasmid designs used in this study for astaxanthin production. Gene organization in each operon are shown in the lefthand boxes while the biosynthesis precursors and products are shown in the righthand boxes. Upstream operons are located either on a plasmid or in the chromosome. Plasmids that form carotenoid intermediates can be transformed together to complete the pathway.

**Figure 3 bioengineering-10-01033-f003:**
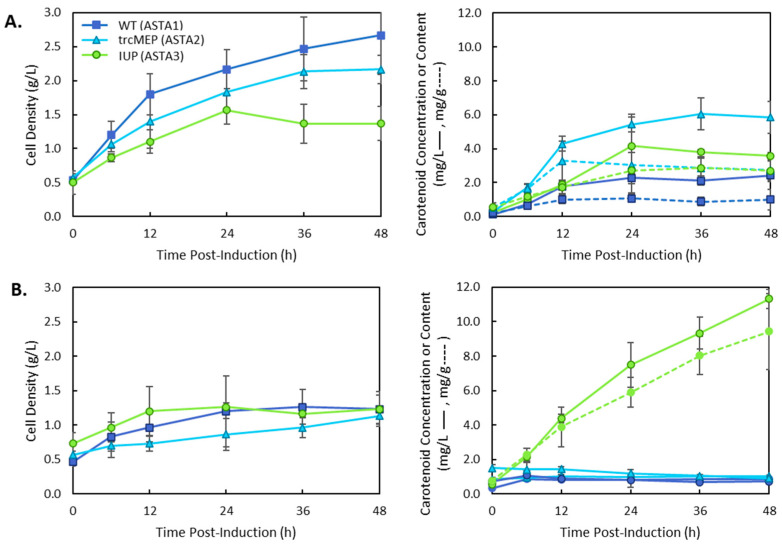
Total carotenoid production in strains 1–3 containing the wild-type (1), trcMEP (2), or IUP (3) pathway and the pAC-BETAipi and pCBFD1 plasmids. Cultures were grown in (**A**) LB media or (**B**) M9 media and induced with 0.1 mM IPTG. Cell growth by dry cell weight is plotted on the lefthand side. Total carotenoids were quantified, and carotenoid concentration (solid lines) and carotenoid content (dashed lines) are shown on the righthand plots.

**Figure 4 bioengineering-10-01033-f004:**
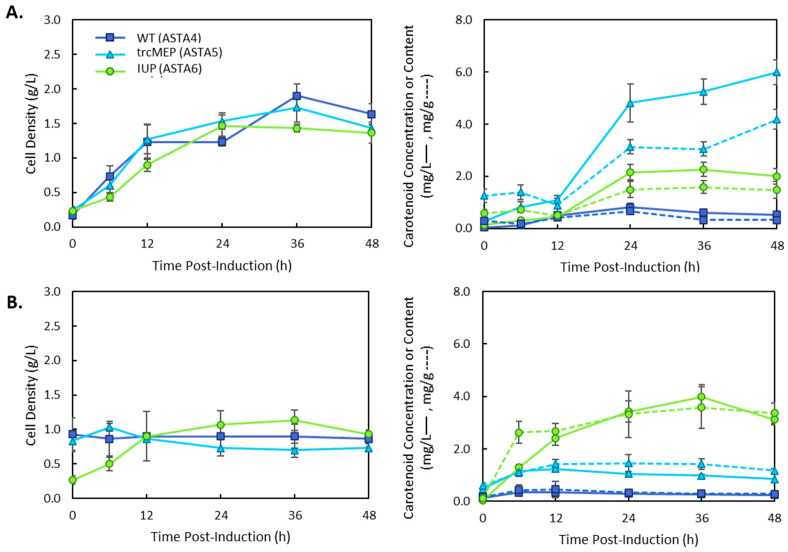
Total carotenoid production in strains 4–6 containing the wild-type (4), trcMEP (5), or IUP (6) pathway and the p5T7-Astaipi plasmid. Cultures were grown in (**A**) LB media or (**B**) M9 media and induced with 0.1 mM IPTG. Cell growth by dry cell weight is plotted on the lefthand side. Total carotenoids were quantified, and carotenoid concentration (solid lines) and carotenoid content (dashed lines) are shown on the righthand plots.

**Figure 5 bioengineering-10-01033-f005:**
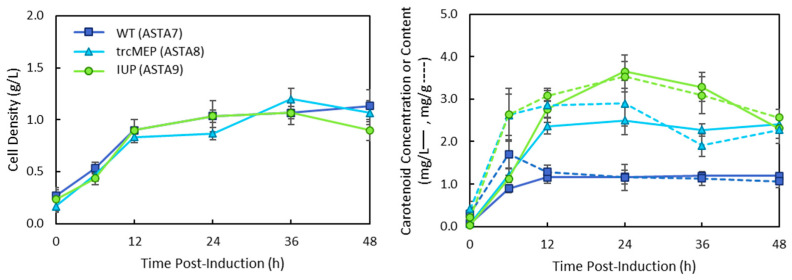
Total carotenoid production in ASTA strains containing the wild-type (7), trcMEP (8), or IUP (9) pathway and the p5T7-lycipi-ispA and pAC-ASTA plasmids. Cultures were grown in M9 media and induced with 0.1 mM IPTG. Cell growth by dry cell weight is plotted on the lefthand side. Total carotenoids were quantified, and carotenoid concentration (solid lines) and carotenoid content (dashed lines) are shown on the righthand plots.

**Figure 6 bioengineering-10-01033-f006:**
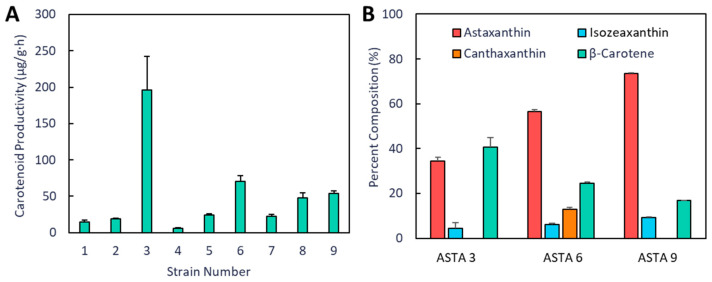
Productivity and carotenoid composition of ASTA strains grown in M9 media. (**A**) Total productivity of each strain over a 48 h cultivation period. (**B**) Percent composition of carotenoids extracted from strains 3, 6, and 9 based on HPLC analysis.

**Figure 7 bioengineering-10-01033-f007:**
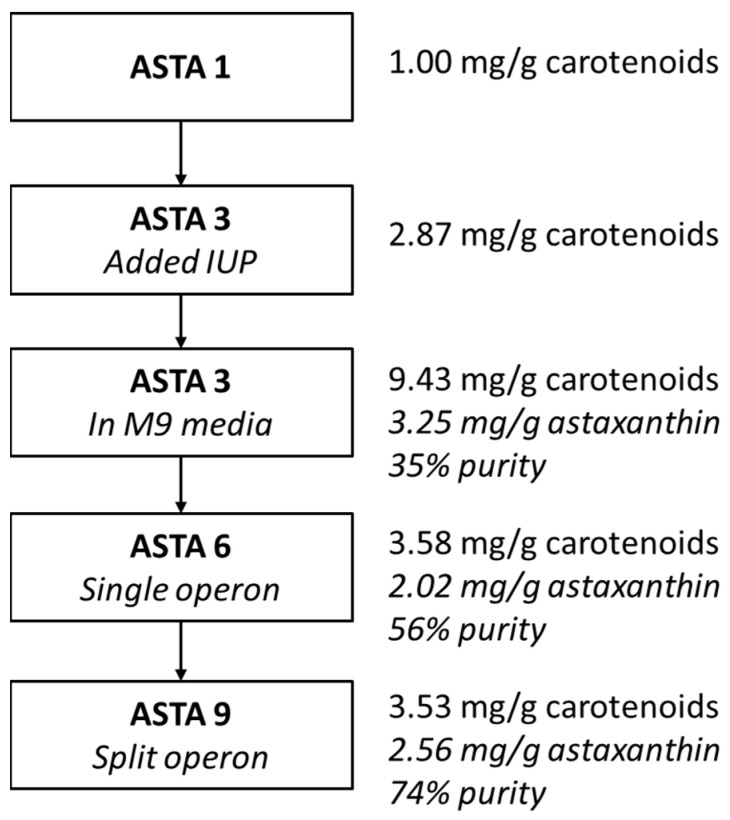
Stepwise improvement of astaxanthin production and purity in the course of this study.

**Table 1 bioengineering-10-01033-t001:** Plasmids and strains used in this study.

Plasmids	Description (ori, Antibiotic Marker, Operon)	Reference
pAC-BETAipi	p15A ori, Cm^R^, P_endogenous_ (*crtE*, *ipi*, *crtY*, *crtI*, *crtB*)	[18]
pCBFD1	pBR322 ori, Ap^R^, P_BAD_ (*cbfd*), P_lac_ (*hbfd*)	[16]
pSEVA228-pro4IUP	RK2 ori, Kn^R^, P_pro4_ (*ck*, *ipk*, *idi)*	[15]
p5T7-lycipi-ggpps	pSC101 ori, Sp^R^, P_T7lacUV_ (*ggpps*, *ipi*, *crtI*, *crtB*)	[15]
p5T7-IspA-ads	pSC101 ori, Sp^R^, P_T7lacUV_ (*ispA*, *ads*)	[15]
p5T7-lycipi-ispA	pSC101 ori, Sp^R^, P_T7lacUV_ (*ggpps*, *ispA*, *ipi*, *crtI*, *crtB*)	This study
p5T7-Astaipi	pSC101 ori, Sp^R^, P_T7lacUV_ (*ggpps*, *crtY*, *cbfd*, *hbfd*, *ipi*, *crtI*, *crtB*)	This study
pAC-ASTA	p15A ori, Cm^R^, P_T7lacUV_ (*cbfd*, *crtY*, *hbfd*)	This study
**Host/Strain**	**Genotype, Plasmids**	**Reference**
MG1655(DE3)	Δ*endA* Δr*ecA* (λ DE3)	[17]
MG1655(DE3)-trcMEP	Δ*endA* Δ*recA* (λ DE3) P_trc_ *dxs-idi-ispDF*	[17]
NEB-5α	*fhuA2 Δ(argF-lacZ)U169 phoA glnV44 Φ80 Δ(lacZ)M15 gyrA96 recA1 relA1 endA1 thi-1 hsdR17*	NEB
ASTA 1	MG1655(DE3), pAC-BETAipi, pCBFD1	This study
ASTA 2	MG1655(DE3)-trcMEP, pAC-BETAipi, pCBFD1	This study
ASTA 3	MG1655(DE3), pSEVA228-pro4IUP, pAC-BETAipi, pCBFD1	This study
ASTA 4	MG1655(DE3), p5T7-Astaipi	This study
ASTA 5	MG1655(DE3)-trcMEP, p5T7-Astaipi	This study
ASTA 6	MG1655(DE3), pSEVA228-pro4IUP, p5T7-Astaipi	This study
ASTA 7	MG1655(DE3), p5T7-lycipi-ispA, pAC-ASTA	This study
ASTA 8	MG1655(DE3)-trcMEP, p5T7-lycipi-ispA, pAC-ASTA	This study
ASTA 9	MG1655(DE3), pSEVA228-pro4IUP, p5T7-lycipi-ispA, pAC-ASTA	This study

Sp^R^ = spectinomycin; Kn^R^ = kanamycin; Ap^R^ = ampicillin; Cm^R^ = chloramphenicol.

## Data Availability

Full plasmid maps are available upon request.

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
