# Peer review of "Production of Astaxanthin Using CBFD1/HFBD1 from *Adonis aestivalis* and the Isopentenol Utilization Pathway in *Escherichia coli"

_bioengineering, 2023, doi:10.3390/bioengineering10091033_

Round 1

Reviewer 1 Report

This manuscript by Roth and Ward dealt with determining the potential of CBFD/HFBD enzyme, from Adonis aestivalis, along with MEP and IUP pathway in production of astaxanthin using E. coli as a host strain. The authors used the properly design experiments to meet the objective of the study. Paper possesses good amount of contents and information. With that being said, some work still needed to be polished to improve the overall quality of paper. The authors are strongly suggested to make proper discussion and make data analysis more solid. Furthermore, there are some unbalanced sentences, grammatical errors and typos giving a bumpy flow to the reading. The authors are strongly encouraged to rewrite the whole manuscript for a cohesive and concise presentation of this interesting work. Thus, the reviewer thought this paper need thorough revision before considering for the publication in Bioengineering.

Major:

Comment 1: The scope and aim of this study is not mentioned clearly. Authors are suggested to clearly state the scope and aim of this study.

Comment 2: In Fig.1, authors indicated the enzyme in two form: (i) Protein name and (ii) Enzyme abbreviation. Authors are suggested to make it uniform (use either protein name or enzyme abbreviation) and to indicate all the genes or protein studied in this study according to Table 1. Authors are also suggested to indicate systematic enzyme name encode by all protein name or enzyme abbreviation in footnotes. Show all the enzyme named mentioned in plasmid.

Comment 3: In case of carotenoid production study using complex media like LB, the production of carotenoids ceased at different time point depending upon the recombinant strain: 12 hours (in case of ASAT1, no MEP or IUP overexpressed), 24 hours (in case of ASAT3, IUP overexpressed) and 36 hours (in case of ASAT2, MEP overexpressed). Interestingly, in case of cultivation in minimal media, there is no or negligible production of carotenoids in case of ASTA1 and ASTA2, but ASAT3 produced highest amount of carotenoid. Similar trend was observed when downstream pathway was overexpressed, there is no or negligible production of carotenoids in case of ASTA4 (no MEP or IUP overexpressed) and ASTA5 (MEP overexpressed), but ASAT6 (IUP overexpressed) produced highest amount of carotenoid. Authors are strongly suggested to describe the observed phenomenon and discuss reason behind it. In addition to this, authors are suggested to show the profile of astaxanthin produced.

Comment 4: Authors overexpressed the downstream pathway (either in single or two plasmid with different gene arrangement) in different background strain (Wild type. MEP overexpressed at chromosome and IUP overexpressed from plasmid). The authors stated that the difference in production of carotenoids is related to difference in protein expression pattern in different construct. If so, authors are suggested to show the SDS-PAGE or mRNA expression profile of each construct. In addition to this, authors are suggested to clarify whether single or multiple ribosomal binding sites are used to overexpress each gene.

Comment 5: Authors are suggested to show the table showing the performance of all strain together in terms of titer, yield and productivity for total carotenoid and astaxanthin.

Comment 6: In this study, authors constructed three plasmid by themselves: named p5T7-lycipi-ispA, p5T7-astaipi-ggpps and pACT7-astaipi. But, there is description of only one plasmid p5T7-lycipi-ispA in table 1, others are missing. Furthermore, authors failed to provide the information regarding to construction of plasmid used in this study. Authors are strongly recommended to provide detail method followed step by step to construct these 3 plasmids: p5T7-lycipi-ispA, p5T7-astaipi-ggpps and pACT7-astaipi. How genes are amplified and inserted at what site of the plasmid.

Comment 7: Authors had made mistake either in naming plasmid or showing genetic organization or both. Different names are given for same plasmid in table, figure and text. The gene arrangement shown in figures and mentioned in tables are completely different. Authors are strongly recommended to rectify it.

Comment 8: In Line 270-271, authors stated “Minimal media supplemented with glucose activates catabolite repression which can lead to lower transcription levels of trc promoter”. Carbon catabolite repression occurred when two different carbon source (glucose and other) are used together. Both T7 and Trc promoter are induced by IPTG. How glucose affect the trc promoter activity. Authors are strongly suggested to clarify the statement.

Comment 9: Authors are suggested to rewrite the abstract dividing into background and scope, methods, results and conclusion.

Minor

Comment 1: Authors are suggested to combine Table 1 and Table 2.

Comment 2: Line 111-16: Authors are suggested to rewrite the sentences. Why transformation was mentioned two times (heat shock and electroporation).

Comment 3: Authors are suggested to change the title “Carotenoid extraction and UV/Vis spectroscopy” to “Quantification of carotenoid” and divide it into two subsection: (i) Carotenoid extraction, (ii) subsection: Quantification: sub-subsection: Spectroscopy, sub-subsection: HPLC.

Comment 4: Authors are suggested to move Fig 6 to supplementary figure.

Moderate editing of english is needed.

Reviewer 2 Report

though title was interesting, need some correction before acceptance

abstract

author should add some numeric findings

author should add some background sentences in introduction section literature review should be done properly that supports the topic.

author need to add objective of study clearly in introduction section

materials and methods

author should add blockdiagram which shows overall work carried out. it seems author scatterd the methodology.

fig.6 and fig 7 not critically discussed author need to add some valuable sentence that supports the both figures

conclusion section is in general perspective need to be revise with the addition of findings in numeric as well as adding suitable sentence 

line 42, line 96 reference missing

English is in the fair form need improvement

Reviewer 3 Report

Abstract

Line 13, 3.2 fold higher content, compared to what?

Lines 16-18, not really clear, please rephrase

Introduction

Line 28, to brighten

Line 39, workhorses

Line 48, reference is missing

Line 57, pyruvate pathway

Materials and Methods

Line 97, ipi?

Line 124, were prepared

Line 136, Table 2

Line 142, to determine the content

Lines 144 and 148, please indicate the centrifugation speed in g instead of rpm (rotor independent)

Line 158, please define MTBE

Results

Line 175, Figure 3

Line 197, to increase

Line 223, Figure 5

Figure 5, part B is missing

Line 234, 2.8 fold increase trc MEP cannot be see in Figure (would it be A or B)?

Line 253, add in caption of Figure 7 that this holds for M9 medium

No specific comments: minor spellcheck required

Author Response

Please see uploaded attachment. 

Round 2

Reviewer 1 Report

This manuscript by Roth and Ward dealt with determining the potential of CBFD/HFBD enzyme, from Adonis aestivalis, along with MEP and IUP pathway in production of astaxanthin using E. coli as a host strain. Authors are still need to clarify some issues in their manuscript.

Comment 1: Authors did not discuss why the production of carotenoid ceased at different time point depending upon the recombinant host strain more clearly (previous comment 3). Authors are suggested to highlight this.

Comment 2: The title of the paper state “Production of astaxanthin……….”. But in all the figure, they only showed the profile of total carotenoid concentration and content. The profile of astaxanthin profile was missing. Authors only showed astaxanthin as the composition percentage of total carotenoid in Fig 6. Authors are encouraged to show the astaxanthin profile or discuss in the text regarding to other strain also as the paper is about the production of astaxanthin specially according to title of paper.

If authors think about the repetition of data in text or figure while showing the titer, yield and productivity of astaxanthin and/or total carotenoid of the strain developed, they are encouraged to put it in supplementary so that reader easily understand how gradually improvement was obtained by seeing only table.

Comment 3: Authors ignored the comments about the construction of plasmids. Authors are suggested to indicate somewhere how this plasmid were constructed for reader understanding.

Comment 4: Its very confusing to follow the naming and genetic organization given in table and figure. The naming and genetic organization given are different in table and figure. For example: for pCBDF1, in table genetic organization is pBAD-cbdf1-T7-hbdf1 but in figure name is pCBDF1/HBDF1 and genetic organization is T7-cbdf1-T7-hbdf1. There are many others like this, authors are strongly suggested to make it uniform.

Comment 5: Yes, carbon catabolite repression is active when glucose is present, its effect will be seen clearly when other carbon sources are present. When glucose (substrate) and arabinose (inducer for PBAD) are added together it could affect the activity of PBAD promoter. For trc promoter, IPTG is used as inducer. As author suggested if glucose affect the trc promoter, it could be indirectly to trc promoter itself. Authors are suggested to highlights what cause glucose affecting trc promoter more clearly.

Minor editing is required

Reviewer 2 Report

Author did all the correction paper now can be accepted for the publication

N/A

Author Response

Thank you.